# H/ACA snR30 snoRNP guides independent 18S rRNA subdomain formation

Paulina Fischer[1,4], Matthias Thoms[2,4], Benjamin Lau [1,3], Timo Denk [2], Maria Kuvshinova[1], Otto Berninghausen [2], Dirk Flemming [1], Ed Hurt [1] ✉ & Roland Beckmann [2] ✉

Ribosome biogenesis follows a cascade of pre-rRNA folding and processing steps, coordinated with ribosomal protein incorporation. Nucleolar 90S pre-ribosomes are well-described stable intermediates, composed of pre-18S rRNA, ribosomal S-proteins, U3 snoRNA, and ~70 assembly factors. However, how numerous snoRNAs control pre-rRNA modification and folding during early maturation events remains unclear. We identify snR30 (human U17), the only essential H/ACA snoRNA in yeast, which binds with Cbf5-Gar1-Nop10-Nhp2 to a pre-18S rRNA subdomain containing platform helices and ES6 of the 40S central domain. Integration into the 90S is blocked by RNA hybridization with snR30. The snoRNP complex coordinates the recruitment of early assembly factors Krr1-Utp23-Kri1 and ribosomal proteins uS11-uS15, enabling isolated subdomain assembly. Krr1-dependent release of snR30 culminates in integration of the platform into the 90S. Our study reveals the essential role of snR30 in chaperoning central domain formation as a discrete assembly unit externalized from the pre-ribosomal core.

Ribosomes, which synthesize the proteins of the cell, consist of a large 60S and small 40S subunit that are built from ribosomal RNA (rRNA) and ribosomal proteins[1]. In eukaryotic cells, ribosome synthesis starts in the nucleolus with the transcription of a poly-cistronic precursor rRNA (called 35S pre-rRNA in yeast) by RNA polymerase I, which is followed by a series of assembly, modification and processing steps, during which roughly 200 ribosome assembly factors (AFs) transiently associate with the evolving pre-ribosomes (for review see refs. 2,3). After export into the cytoplasm, the final maturation steps take place[4]. One of the first stable assembly intermediates formed during ribosome construction is the 90S pre-ribosome[5–7]. This intermediate was isolated by affinity-purification from *Saccharomyces cerevisiae* (yeast)[8], *Chaetomium thermophilum*[7] and human cells[6], and structure determination by cryo-electron microscopy (cryo-EM) showed how the ~70 different 90S AFs, often organized in modules, constitute a scaffold around the nascent 18S pre-rRNA, which consists of the 5′, central, 3′ major

and 3′ minor domains. These domains each recruit a specific set of assembly factors co-transcriptionally that support rRNA maturation and domain integration into an increasingly compact 90S pre-ribosome (Supplementary Fig. 1a)[5,7–11]. In addition to the protei-naceous components, the 90S carries a single small nucleolar RNA, the U3 snoRNA[5,12,13] belonging to the group C/D box ribonucleo-protein particles (RNPs), which usually perform 2′-O-ribose methylation[14–16]. However, U3 is not active in snoRNA-guided pre-rRNA modification but together with its core factors Nop56, Nop58, Snu13 and Nop1 performs a structural role in the 90S by locally hybridizing to the 5′ external transcribed spacer (ETS) and 18S rRNA, thereby keeping the nascent rRNA organized but still immature[17–19]. In contrast, the majority of the other existing snoR-NAs (about 75 in yeast and hundreds in humans) perform specific pre-rRNA modifications during ribosome biogenesis, mostly located in functionally important regions in both ribosomal subunits[20–22]. Here, H/ACA snoRNAs and their protein core factors,

[1]Biochemistry Center, Heidelberg University, Heidelberg, Germany. [2]Department of Biochemistry, Gene Center, University of Munich, Munich, Germany. [3]Present address: Molecular Systems Biology Unit, European Molecular Biology Laboratory (EMBL), Meyerhofstrasse 1, Heidelberg, Germany. [4]These authors contributed equally: Paulina Fischer, Matthias Thoms. ✉e-mail: ed.hurt@bzh.uni-heidelberg.de; beckmann@genzentrum.lmu.de

the enzymatic subunit Cbf5 and Nhp2-Gar1-Nop10, mediate pseudouridylation[16,23].

Structural information on H/ACA RNPs stem mainly from crystal structures of recombinant archaeal[24] complexes, yeast H/ACA factors[25] and more recently, cryo-EM structures of human telomerase, which is tethered to an H/ACA RNP complex by telomerase RNA[26–28]. Yet, the function of AFs and snoRNPs acting early in 90S assembly remained largely enigmatic. The myriad of snoRNAs are thought to function in the earliest, co-transcriptional phase of 90S formation, when the guide snoRNAs may still reach the cognate nucleotides on the primary transcript for hybridization and site-specific modification, before they become inaccessible in the later compactly folded rRNA. Intriguingly, only 3 of the snoRNAs involved in ribosome biogenesis are essential in yeast (U3[29], U14[30], snR30[31–33]) and two of them, U3 and snR30, are not active in modifying rRNA but have evolved to perform other important functions instead. With the function of the C/D box snoRNA U3 identified in structurally coordinating the 90S, we set out to elucidate the poorly understood essential function of the H/ACA snR30 snoRNA. Here, we report the cryo-EM structure of the H/ACA snR30 snoRNP, showing previously unresolved architectural and mechanistic principles of snR30 function during early 90S formation.

## Results

### Role of Krr1 for H/ACA snR30 function

We sought to investigate the role of the 90S AF Krr1, since it is associated with a distinct 18S rRNA domain, which comprises rRNA helices h20 to h23 and expansion segment 6 (ES6), the latter being targeted by the H/ACA snoRNA snR30[6–9,34] (Fig. 1a). Moreover, Krr1 physically and genetically interacts with the early 90S factor Kri1 that is also known to bind snR30[35,36]. Krr1 is composed of two essential KH domains and acts as placeholder for the AF Pno1/Dim2 on the maturing 90S at a site, which eventually constitutes the 'platform' of the mature 40S subunit[37]. Finally, the 40S platform together with the 40S head forms the exit channel for the mRNA on the translating ribosome. Besides the KH domains, Krr1 carries a non-essential C-terminal α-helix that contacts the 5′ domain of the 18S rRNA and the Kre33 module in the 90S[7] (Fig. 1a, b). Additionally, a highly conserved sequence of ~80 amino acids in length in the C-terminal part connects the Krr1 KH domain core and C-terminal α-helix, the function of which remains elusive. We termed a region within this sequence **p**latform **b**inding **m**otif (**PBM**) (Figs. 1a–c; and Supplementary Fig. 2), which exhibits a highly conserved **p**roline-**r**ich **m**otif between residues 248–257 (**PRM**; Figs. 1b, c; and Supplementary Fig. 2), followed by a short α-helix (residues 258–267). Here, we discovered that this intervening sequence performs an essential function (krr1ΔC3 or Δ248–317) and its gradual truncation (krr1ΔC1 or Δ268–317 to krr1ΔC2 or Δ258–317) causes a progressive growth inhibition (Fig. 1c; Supplementary Fig. 3a). In addition, overexpression of the krr1ΔC3 construct in the presence of wild-type KRR1 induced a dominant-negative phenotype, suggesting that Krr1 lacking the PBM can assemble into the 90S but subsequently inhibits further 90S biogenesis (Fig. 1d). In contrast, deletion of the Krr1 KH domains (krr1ΔN), although lethal in itself, did not cause a growth defect upon overexpression (Fig. 1c, d). Thus, the conserved PBM between the Krr1 KH domain and the C-helix performs a vital function during 90S biogenesis.

To obtain a functional understanding of Krr1's PBM during different phases of 90S formation, we affinity-purified early (Utp10-FTpA or Utp7-FTpA) and late 90S particles (Noc4-FTpA) from the different krr1 C-terminal deletion mutants and analyzed them by mass spectrometry (MS) for their composition (Figs. 1e, f and Supplementary Fig. 1b, c). Early 90S particles were characterized by the presence of AFs acting during early 90S formation (e.g., Mrd1-Nop9-Nsr1)[38,39], while intermediate factors joining before the 90S compaction (e.g., Utp20, Kre33) and dual 90S/40S factors (e.g., Dhr1, Dim1) marked a fully assembled, late 90S pre-ribosome[9–11] (Supplementary Fig. 1a). First, we

over-expressed the dominant-negative krr1ΔC3 allele lacking the entire PBM as well as wild-type KRR1 for a mock control and isolated late 90S via Noc4-FTpA from the respective whole cell lysates. Comparison of the final eluates showed a similar 90S factor enrichment, but incorporation of the UTP-C complex (Utp22-Rrp7) was specifically reduced in the case of the krr1ΔC3 mutant (Fig. 1e, f). Within the conserved C-terminal region, the PRM makes a contact in the 90S structure with the ribosomal protein uS15 (Rps13) (Fig. 1a). Concurrently, uS15 interacts with a short C-terminal helix of Rrp7, which in turn interacts with Utp22 to form the essential UTP-C module on the 90S (Fig. 1a). Thus, not only the Rrp7 C-terminus is crucial for UTP-C recruitment to the 90S as previously shown[40], but also Krr1 via its proline-rich motif might participate in this process[37]. Moreover, we observed reduced levels of the RNA helicase Rok1 and its cofactor Rrp5 in particles isolated from mutant cells (Fig. 1e, f). Rok1 is a DEAD-box RNA helicase implicated in snR30 removal, with Rrp5 described as its co-factor required for Rok1 recruitment[41–43]. Unexpectedly, Utp23 and Kri1, two factors that were suggested to act transiently in earlier steps of the 90S pathway[9,10], were strongly enriched in the krr1ΔC3 derived precipitate (Fig. 1f). Since Utp23 was also described to be required for snR30 release, these findings suggested that Krr1 might as well be involved in this process.

Next, we affinity-purified 90S via Noc4-FTpA and Utp10-FTpA, from the viable but slow growing krr1ΔC1 strain carrying only a partial deletion in the Krr1 PBM (Supplementary Fig. 1c). This revealed, similar to what we found for the krr1ΔC3 mutant, a specific decrease of UTP-C and Rrp5-Rok1, and an increase of Utp23-Kri1 in comparison to the wild-type strain (Fig. 1f). Strikingly, also H/ACA snoRNP core factors Cbf5-Gar1-Nop10-Nhp2, but not C/D box core factors were elevated (Fig. 1f). This data suggests that the different krr1 PBM mutants specifically affect the dynamic association of a few 90S AFs with the developing 90S, amongst which are Utp23-Kri1 and H/ACA core factors. To further investigate how these factors become trapped on 90S particles upon manipulating the Krr1 platform binding motif, we affinity-purified Utp23-FTpA and Kri1-FTpA. As anticipated, both baits strongly co-precipitated H/ACA and C/D box factors, but typical 90S factors were present only sub-stoichiometrically (e.g., Utp20, Rrp5, Utp22). In contrast, when Utp23 or Kri1 were isolated from the krr1ΔC1 strain, both proteins associated with 90S particles enriched in late 90S AFs, but UTP-C and Rrp5-Rok1 that interact with the 18S central domain still remained low abundant (Supplementary Fig. 3b, c). This data corroborates that the release of Utp23-Kri1, which are implicated in snR30 interaction[36], from the developing 90S requires an intact Krr1 platform binding motif. Thus, the halted 90S maturation observed in the krr1 PBM mutants may be caused by a failed release of the H/ACA snR30 snoRNP, which in consequence alters the dynamic association of a few other 90S biogenesis factors that bind to the 40S platform region.

To directly test whether the association of snR30 with 90S particles is affected by the krr1ΔC1 mutation, we isolated early 90S pre-ribosomal particles via Utp7-FTpA and late 90S via Noc4-FTpA (see above). The final eluates were analyzed by Northern blotting, which barely detected snR30 in wild-type (late) 90S purified via Noc4-FTpA, but snR30 was clearly enriched when the same bait was isolated from the krr1ΔC1 mutant (Fig. 1g). As expected, the distinct presence of snR30 in Utp7-precipitated (earlier) 90S particles was not affected by the krr1ΔC1 mutation, while the association of the essential C/D box snoRNAs U14 and U3 were not influenced by the krr1ΔC1 mutation in neither 90S population (Fig. 1g). Thus, the Krr1's PBM plays a specific role in snR30 release from the early 90S pre-ribosome.

### Structure of the snR30-90S particle

For structural investigations, we aimed at isolating early and late 90S intermediates containing the snR30 snoRNP with increased yield and homogeneity. Therefore, we employed split-tag affinity purifications, using Utp23 as first and wild-type or mutant Krr1 as second bait. Wild-

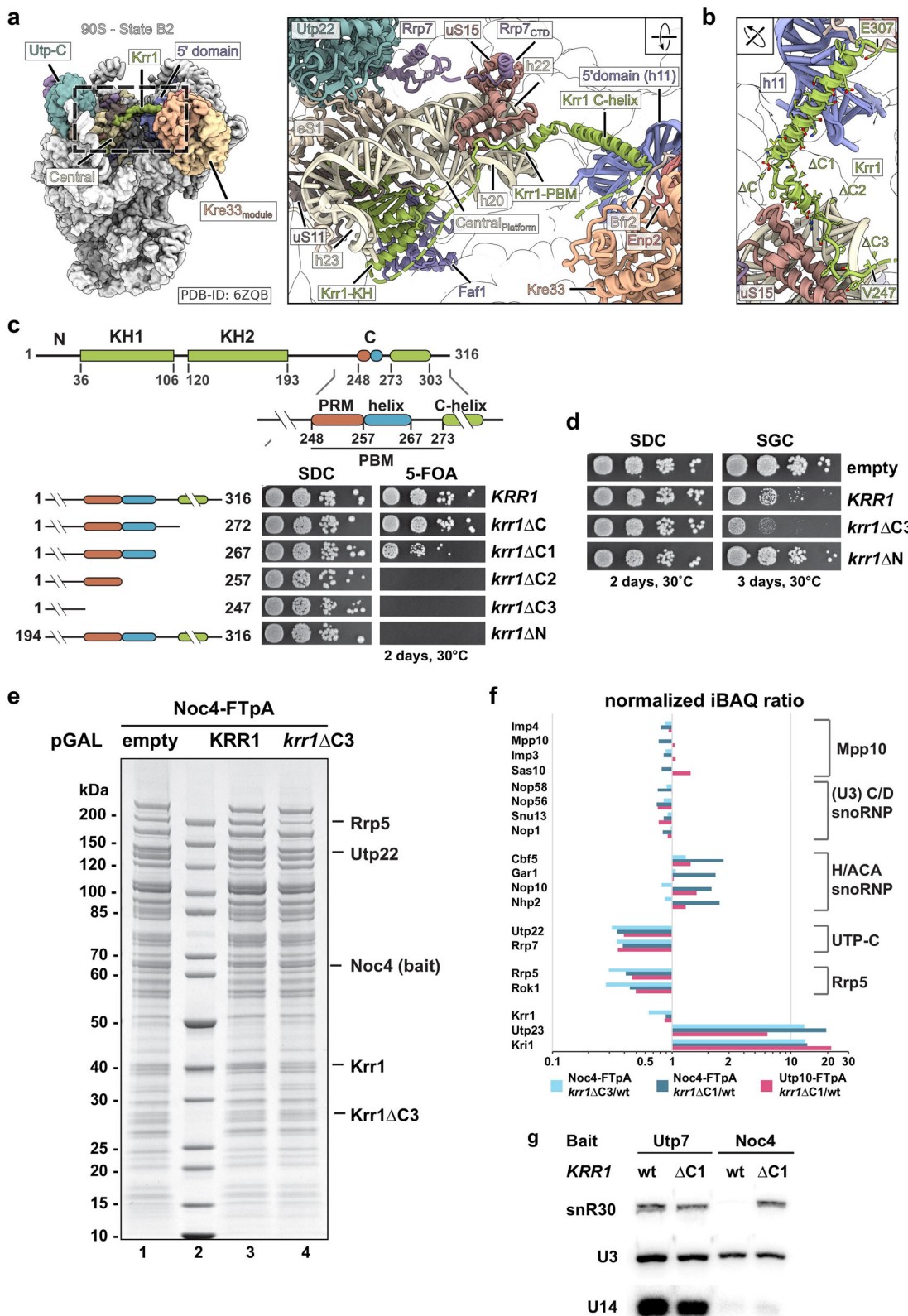

type baits effectively precipitated early 90S particles, indicated by the enrichment of both H/ACA core factors and typical 90S 'upstream factors' (e.g., Mrd1-Nop9-Nsr1) (Fig. 2a, b). In contrast, Utp23 together with mutant Krr1ΔC3 co-precipitated 90S particles that were progressed in maturation (increased levels of intermediate 90S factors, e.g., Kre33 and Noc4 modules, and decreased levels of early 90S factors), but H/ACA factors were still present (Fig. 2a, b). Sucrose gradient

centrifugation confirmed that the wild-type Utp23-Krr1 preparation contained early 90S particles (e.g., Mrd1 and Nop9) (Fig. 2c, d). However, we noticed a second slower migrating complex on the top of the sucrose gradient, which contained snR30, H/ACA factors Cbf5-Gar1-Nhp2, Kri1-Utp23, and ribosomal proteins uS11 and uS15 (Fig. 2c, d). In contrast, the preparation using Utp23-Krr1ΔC3 showed a further developed 90S that lacked e.g., Mrd1 but contained Dhr1 and other

**Fig. 1 | Timely release of snR30 snoRNP depends on Krr1's conserved proline-rich motif. a** Overall structure of a pre-A₁ 90S particle (state B2, PDB: 6ZQB) with Krr1 highlighted in the central domain and its interaction network including uS11, uS15, h20-23 of the 18S central domain and Rrp7 via uS15. **b** Boundaries of the progressive C-terminal truncation mutants of Krr1 and **c** complementation assay including a deletion of the KH domains of Krr1 (krr1ΔN). **d** Galactose-induced (SGC) overexpression of Krr1 variants compared to growth without induction (SDC). **e** SDS-PAGE of intermediate 90S pre-ribosomes isolated via AF Noc4 from cells grown in galactose containing medium containing either an empty plasmid (lane 1), galactose-inducible wild-type *KRR1* (lane 3) or *krr1ΔC3* (lane 4). Protein bands identified by MS are labeled accordingly. **f** SemiQ-MS analysis of 90S pre-ribosomes isolated via Utp10 or Noc4 from wt or mutant cells. Intensity-based iBAQ values were normalized to the UTP-B factor Pwp2, and log₁₀ of ratios (mutant/wt) are shown according to their x-fold change. Selected proteins are grouped according to their 90S biogenesis modules shown on the right. **g** Northern blot analysis of snoRNAs in early (Utp7) and intermediate (Noc4) 90S pre-ribosomes from wild-type and *krr1ΔC1* cells using specific probes targeting snR30, U3 and U14. Experiments were performed more than twice (**c d**), twice (**e**) or once (**f g**).

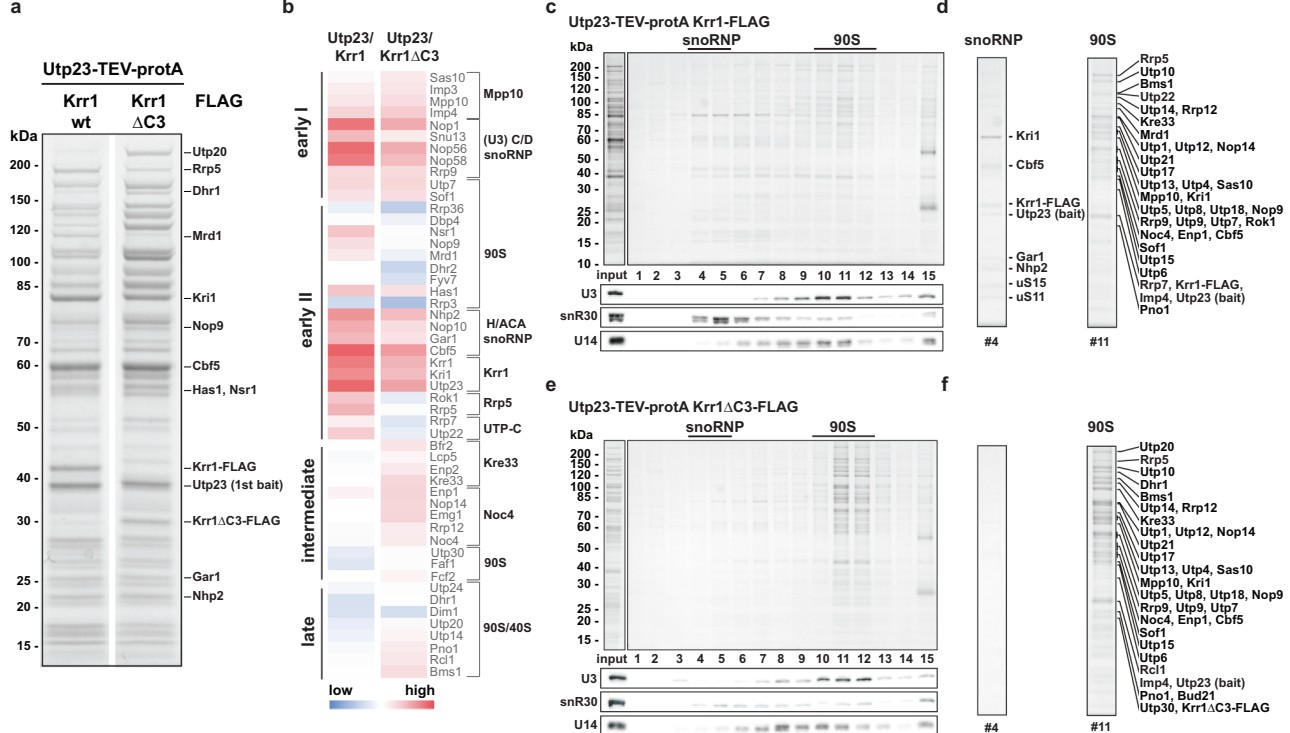

**Fig. 2 | snR30 accumulation is observed on a stalled 90S pre-ribosome containing the Krr1ΔC3 mutant. a** SDS-PAGE and **b** heat map of proteins identified in the final eluates from split-tag purifications of Utp23-Krr1 and Utp23-Krr1ΔC3. Intensity-based iBAQ values were normalized to the UTP-B factor Pwp2, and log₁₀ values were colored from low (blue) to high (red). Identified proteins are grouped according to their 90S biogenesis module on the right, or according to their dynamic association (Supplementary Fig. 1a) on the left. **c** SDS-PAGE and northern blot analysis of split-tagged Utp23-Krr1 particles fractionated on a sucrose gradient. **d** H/ACA core snoRNP components as well as 90S AFs are labeled as identified by MS. **e** SDS-PAGE and northern blot analysis of sucrose gradient fractionations of split-tagged Utp23-Krr1ΔC3 particles (left panel). **f** 90S AFs are labeled as identified by MS. Experiments were performed more than twice (**a**) and once (**b–e**).

typical late 90S factors, as well as reduced levels of the slower migrating snR30 complex (Fig. 2e, f). These observations confirmed that the 90S continued maturation in the case of Krr1ΔC3 but failed to release Utp23-Kri1 and the snR30 snoRNP. Particles isolated via Utp23-Krr1ΔC1 (Krr1ΔC1 carries a partial deletion of the PBM), however, were separated into a mixture of early and late 90S (Supplementary Fig. 3d, e). Thus, the progressive growth defect of the Krr1 C-terminal truncation mutants is mirrored by the ability to release the snR30 snoRNP from early 90S particles. Moreover, the Krr1 mutant 90S particles indicate the time point during 90S maturation, at which the block of snR30 release leads to lethality.

Based on our biochemical data we performed cryo-EM analysis of the two different snR30 containing 90S particles that were obtained with wild-type Utp23-Krr1 and mutant Utp23-Krr1ΔC3 split baits. However, our attempts to structurally characterize the early 90S from the wild-type combination were limited due to aggregation and a low number of identifiable particles (Supplementary Fig. 4a–c). In contrast, the structural analysis of the Utp23-Krr1ΔC3 isolate, that yielded more progressed but stalled 90S, revealed two distinct, but tethered particle populations: one was identified as the snR30 snoRNP complex

(Figs. 3a–c and Supplementary Fig. 4–6, Supplementary Table 1, Supplementary Movie 1), while the other exhibited a series of stalled 90S states. When we extracted the snoRNP particles with an enlarged box size and performed 2D classification, we identified a large but diffuse density adjacent to the snR30 snoRNP population, which likely resembles the 90S core particle. (see below and Figs. 3a, Supplementary Fig. 4d–f and 6d, Supplementary Table 1).

Notably, the particle population with the snR30 snoRNP is not a detached free complex, but part of two rigid, connected modules: one module representing the H/ACA snoRNP core that is composed of snR30 snoRNA, two copies of the canonical H/ACA constituents Cbf5, Nop10 and Nhp2 but only one copy of Gar1 (Fig. 3b, c). The second module is part of the later 90S, but here found externalized from the monolithic 90S core, as it carries the 18S rRNA platform helices h20, h22-h23, the ribosomal proteins uS15 and uS11 and in addition three specific 90S AFs, Krr1, Kri1 and the globular N-terminal PIN domain of Utp23 (Fig. 3b, c, Supplementary Movie 1). The model of the Krr1ΔC3 mutant includes the N-terminus, the KH domains and the C-terminus up to F227 (Fig. 3b, c) and hence ends upstream of the applied truncations. Importantly, within this higher order complex at a resolution

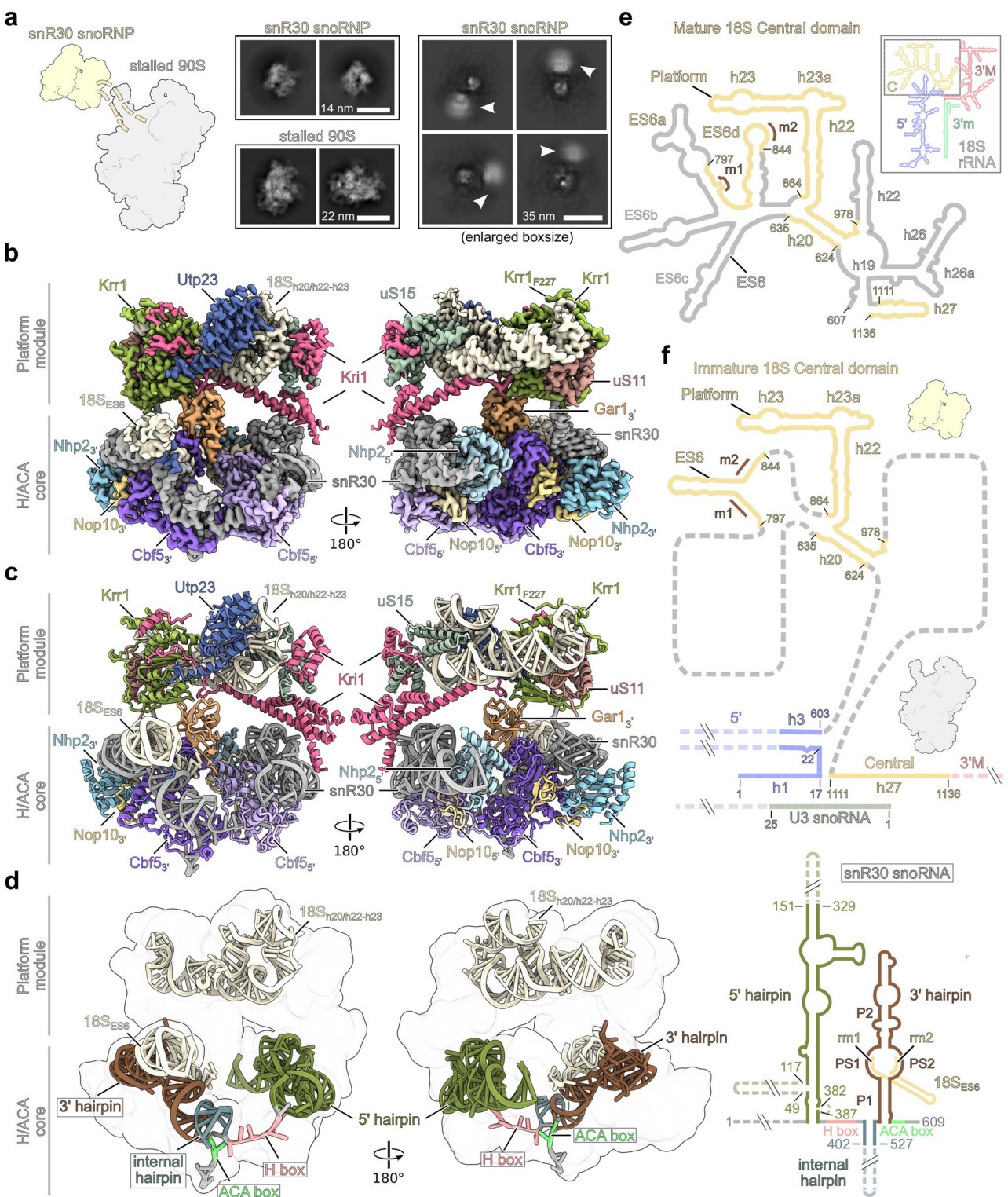

of 2.9 Å, parts of the snR30 5' RNA hairpin and the complete essential 3' hairpin can be well traced (Fig. 3d and Supplementary Figs. 5c and 7a), with the latter hybridizing with a region of expansion segment ES6 of the 18S rRNA (nucleotides 797 to 844) (Fig. 3d–f). With 609 nucleotides in length, the yeast snR30 has an unusually large size compared to its homologs (e.g., 207 nucleotides of human U17)[31,32]. The drastic difference in size emerges mainly from the non-essential extended 5' and internal hairpin in yeast, which were unresolved in our snoRNP structure, likely due to flexibility (secondary structure in Fig. 3d). The complementary 90S core structures largely resembled several late 90S

states with the latest being most similar to the previously observed 90S states B2, preceding $A_1$ cleavage[8]. Interestingly, however, the 90S pre-ribosomes displayed a complementary gap at the site where the ES6 and 40S platform module of the central domain would be integrated (see below and Supplementary Fig. 4f). This finding further supports our interpretation that these stalled 90S intermediates carry a flexibly tethered snR30 snoRNP complex that itself is bound to the not yet incorporated immature 18S rRNA platform (Fig. 3a).

To verify that the bipartite organization of the snR30 snoRNP with H/ACA core and Krr1ΔC3 platform module is representative of

**Fig. 3 | Structure of the snR30 snoRNP. a** Schematic of the flexibly connected snR30-90S particle (left panel), 2D class averages of the snR30 snoRNP and stalled 90S particles purified via Utp23-Krr1ΔC3 (middle panels) and 2D class averages of the snR30 snoRNP from particles extracted with an enlarged box size (right panel). The large fuzzy density next to the snR30 snoRNP is highlighted with an arrow and may represent the associated 90S particle. Scale bars are shown. **b, c** Cryo-EM density map (**b**) and molecular model (**c**) of the snR30 snoRNP purified via Utp23-Krr1ΔC3 (State 1) depicted in two orientations. The bipartite assembly consists of the H/ACA snoRNP core and the rigidly bound platform module. The factors are colored and labeled. The C-terminal end of the Krr1 model is labeled with Krr1 F227. **d** Molecular model of the snR30 snoRNA and 18S rRNA within the snR30 snoRNP shown in the same orientations as in (**b, c**), and secondary structure organization of the snR30 snoRNA including hybridization of the snR30 3′ hairpin with 18S-ES6 rRNA (right panel). The individual structural elements are colored and highlighted.

Regions not covered by the molecular model are indicated by dashed lines. **e** Secondary structure of the mature 18S rRNA highlighting the 18S domains: 5′, C (Central), 3′M (3′ major) and 3′m (3′ minor) (upper right panel) and close-up of the mature central domain (left panel). rRNA regions identified within the snR30 snoRNP model and the nucleotides of 18S rRNA h27, which are part of the stalled 90S, are colored in yellow. Hybridization sites with the snR30 3′ hairpin are indicated in brown (m1, m2). **f** Secondary structure of the 18S rRNA within the snR30 snoRNP and the stalled 90S particles. The ES6 forms an alternative helix compared to the mature organization (see **e**). Regions of the rRNA not included in the model are shown as dashed gray lines. For rRNA secondary structures, the 18S rRNA structure provided by http://apollo.chemistry.gatech.edu/RibosomeGallery/ was used and modified. Released under a Creative Commons Attribution-ShareAlike 3.0 license (**e, f**).

intermediates contained in early 90S particles that are formed under physiological conditions, we made an effort to structurally analyze the wild-type Utp23-Krr1 split-bait preparation (Fig. 2a, c). Even though it was not possible to identify structurally distinct early 90S classes in the Utp23-Krr1 wild-type preparation, likely due to the limited compaction of these early intermediates, we indeed found essentially the same modular architecture of the snR30 snoRNP core and platform modules albeit resolved at lower resolution (Supplementary Fig. 4c, 6a). Thus, our biochemical and structural data indicate that the snR30 snoRNP-90S complex is already formed during the initial phase of 90S assembly, representing the physiological phase of snR30 activity.

## Structure of the snR30 H/ACA snoRNP core

The H/ACA core of the snR30 snoRNP is built around the Cbf5 dimer, which is the pseudouridine synthase within all H/ACA snoRNPs. Each monomer binds the snR30 'H' or 'ACA' box consensus site, respectively, and thereby provides the overall orientation of the Cbf5-Nop10-Nhp2 tandem (Figs. 4a, b and Supplementary Fig. 7b). The two H/ACA modules in turn coordinate the 5′ and 3′ snoRNA hairpins, which each display a ~90 degree kink and form part of the Nhp2 interaction patch (Fig. 4c). Unexpectedly, in our snoRNP core structure, Gar1 interacted exclusively with Cbf5 in the 3′ module and a second Gar1 copy is prevented from binding to the 5′ module due to a potential steric clash between its interaction partner Cbf5 and an extended region of the snR30 RNA (Fig. 4d). Importantly, the overall architecture of each of the two modules of the snR30 snoRNP resemble that of the archaeal H/ACA complex[44] and the whole snR30 tandem is highly similar to the human H/ACA snoRNP counterpart recently characterized in the human telomerase, where it has a function in biogenesis without any known modification activity[45] (Supplementary Fig. 7b). Thus, the organization of H/ACA snoRNPs has remained structurally conserved from Archaea to humans, even for complexes, which are no longer active in pseudo-uridylation as is the case for both the yeast snR30 snoRNP (see below) and human telomerase[27,28].

## Pseudo-uridylation dormant snR30 snoRNP

In the 3′ module of the H/ACA snoRNP core, we identified the base pairing of the snR30 with its 18S rRNA target sequence involving two conserved motifs within ES6, called **rm1** and **rm2** (also **PS1** and **PS2**, for **p**seudo-uridylation guide **s**equence **1** and **2**), largely as predicted previously[34] (Figs. 3d–f and 4e, f). With this interaction, a premature 18S rRNA conformation is supported, including a stable rRNA helix (nucleotide 806–836) forming a three-way junction with the snR30:18S hybrid region (Fig. 4e, f; see also ref. 46). This immature rRNA helix then becomes part of ES6a and ES6d in the mature 18S rRNA conformation (Fig. 3e, f). The observed mode of the snR30:18S interaction is similar to the productive conformation observed before in archaea (with PS2 providing most of the interaction surface with Cbf5). However, the two hybrid helices PS1 (rm2) and PS2 (rm1) are located in the basal part of the pseudo-uridylation pocket near the P1 stem, while

productive hybrid helices would be located in the distal part near the P2 stem (Figs. 4e, f and Supplementary Fig. 7b). Moreover, instead of placing a short loop between PS1 and PS2 and offering a potential modification target, the 18S rRNA stretches away from the active site to connect with the rRNA h22 of the platform module and the unresolved parts of ES6. The connection with h22, however, was only observed at low resolution and at lower contour levels due to apparent flexibility (Supplementary Fig. 7c). Within the archaeal H/ACA RNP complex, the substrate RNA is stabilized in the active site by a conserved thumb loop of Cbf5 in a closed conformation. In the 3′ H/ACA module of our structure, this loop adopted an almost identical conformation. However, with the 18S rRNA on top of it, the thumb loop is shielding it from the active site of Cbf5 (Fig. 4g). In the 5′ module of the snoRNP core, the Cbf5-Nop10-Nhp2 trimer binds exclusively to the snR30 RNA which mimics a PS2 helix (Supplementary Fig. 7d). Since this helix simply continues without any potential modification site being exposed, the active site of the 5′ Cbf5 is sterically blocked and the thumb loop is completely delocalized, lacking the Gar1 protein (Fig. 4d). A similar unproductive mode was observed for the human telomerase associated H/ACA RNP (Fig. 4g). These different structural arrangements explain the inability of snR30 to pseudouridylate the 18S rRNA within the context of the 90S pre-ribosome.

## Organization of the snR30 snoRNP-platform module

A further inspection revealed that the AF's Kri1 and Krr1 together with Gar1 clamp the H/ACA core and the platform modules together, therefore stabilizing the overall rigid bi-lobed structure (Fig. 5a, b). Here, three major interactions can be seen: first, direct Kri1 binding to snR30 via interactions with the bulged-out 5′ hairpin RNA bases A129 and G363; second, Krr1 directly interacting with U565 and A566 of the snR30 3′ hairpin, where the protruding U565 inserts into a Krr1 pocket, while R86 of Krr1 establishes a cation-pi stacking with A566 of the snR30; and third, Kri1 and Krr1 interacting with the single 3′ Gar1 copy of the snoRNP module (Fig. 5a). In canonical H/ACA snoRNPs, Gar1 enhances Cbf5 activity and facilitates product release[47], however, in our structure the single Gar1 copy has an alternative function in supporting the snoRNP-platform module structure. Moreover, the snR30 snoRNP is rigidly interwoven with the 18S rRNA platform helices h20, h22 and h23 that already recruited their dedicated ribosomal proteins uS15 and uS11 in a close-to-mature conformation (Fig. 5b and Supplementary Fig. 8). The platform helices are contacted by the PIN domain of Utp23 which in addition also interacts with Kri1 and Krr1. The latter two interact via the N-terminus of Kri1 and the KH domain of Krr1 as suggested previously[35] (Fig. 5b). However, the 90S AFs Kri1 and Krr1 are also associated with uS15 and uS11 and thus become tightly integrated into this higher order assembly (Fig. 5b). Specifically, uS15 is wedged between two α-helical parts of Kri1 and uS11 is decorated by Krr1's KH domains and its meandering N-terminus (Fig. 5b). This structural arrangement is consistent with functional data, which showed that uS15 and uS11, like snR30, are required for the earliest 35S

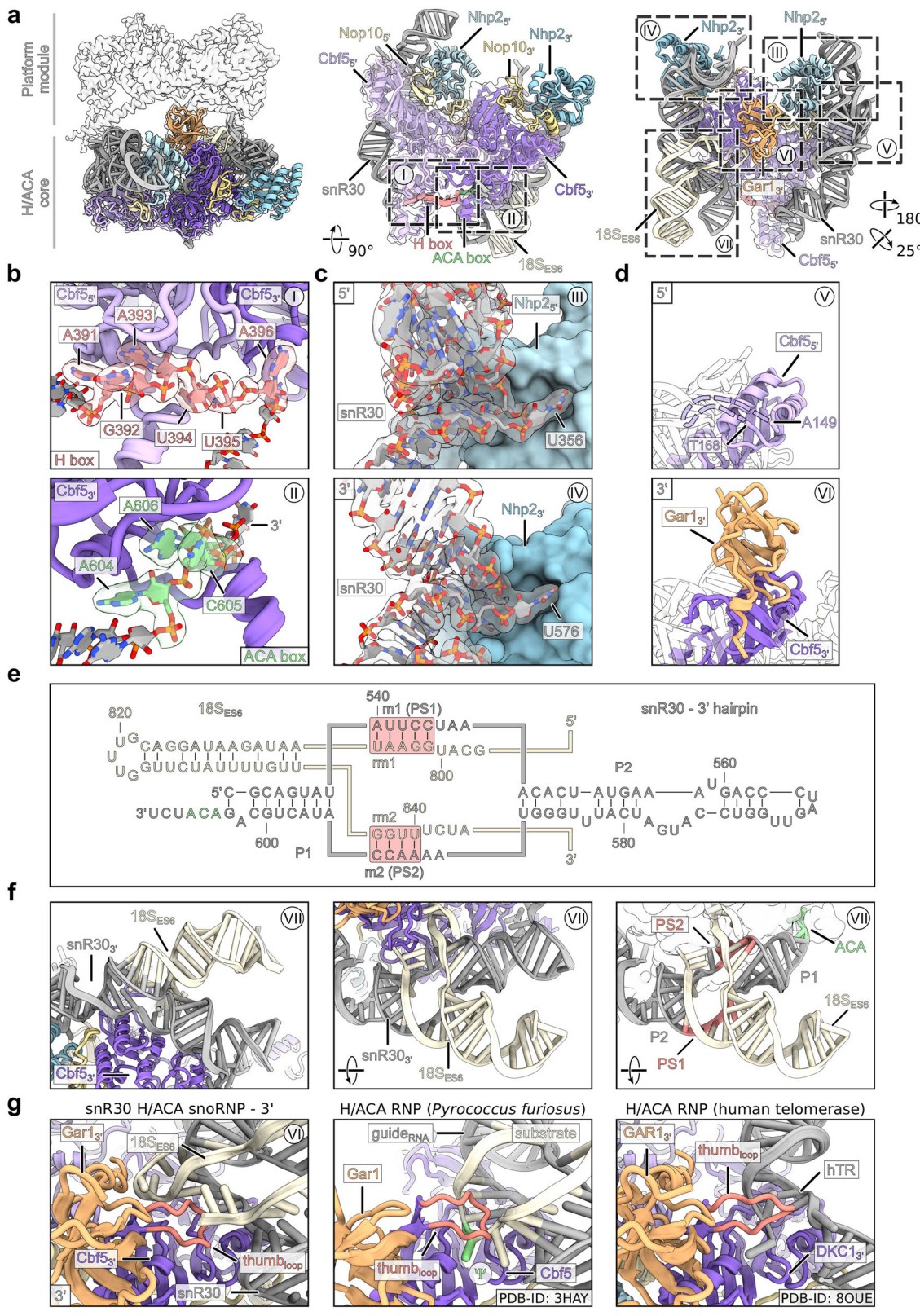

pre-rRNA processing steps in yeast[33,48]. Notably, depletion of uS11 and uS15 in yeast also caused a decreased level of UTP-C and elevated levels of late assembly factors in arrested 90S pre-ribosomes[49]. This phenotype is reminiscent of what we found in this study upon deletion of the Krr1 PBM. Thus, Kri1 and Krr1 safeguard the assembly of uS11 and uS15 into the early 90S pre-ribosome, which explains their early role in 40S subunit biogenesis.

We observed an additional flexible connection linking the H/ACA snoRNP core and the platform module in our complex provided be the AF Utp23 (Fig. 6a). Its unstructured C-terminus recognizes and binds to the snR30-ES6 three-way junction within the 3′ H/ACA module through a conserved motif (Fig. 6b, c). Concurrently, its N-terminal PIN domain is tightly packed against rRNA helices h20, h22 and h23a in the platform module, while its essential and highly positive charged α-helix 1

**Fig. 4 | Structural analysis of the snR30 H/ACA core. a** Overview of the H/ACA module in different orientations. Factors are labeled and regions covered in (**b**–**g**) are highlighted with dashed boxes. **b** Enlarged views of the snR30 H box (top panel) and ACA box (bottom panel) consensus sites interacting with 5' and 3' Cbf5's, respectively. Atomic models and segmented cryo-EM density maps are shown for snR30. **c** Close-up views of the snR30 5' and 3' hairpins interacting with their respective Nhp2 copy. **d** Focus on the two Cbf5 copies in the 5' and 3' halves of the H/ACA core. Only one copy of Gar1 is observed in the snR30 snoRNP structure bound to the H/ACA 3' half. **e** Secondary structure of the snR30 3' hairpin and interaction with the 18S-ES6. The P1, P2 and PS1, PS2 helices are labeled and highlighted. **f** Magnification of the snR30-ES6 three-way junction in different views. **g** Focus on the putative substrate binding region of the 3' half of the H/ACA snR30 snoRNP (left panel) and comparison with the archaea H/ACA RNP from *Pyrococcus furiosus* (PDB-ID: 3HAY) and the human telomerase H/ACA RNP (PDB-ID: 8OUE) (middle and right panels). The modified nucleotide within the archaea H/ACA RNP is highlighted (Ψ) and the Cbf5/DKC1 thumb loops are shown in red.

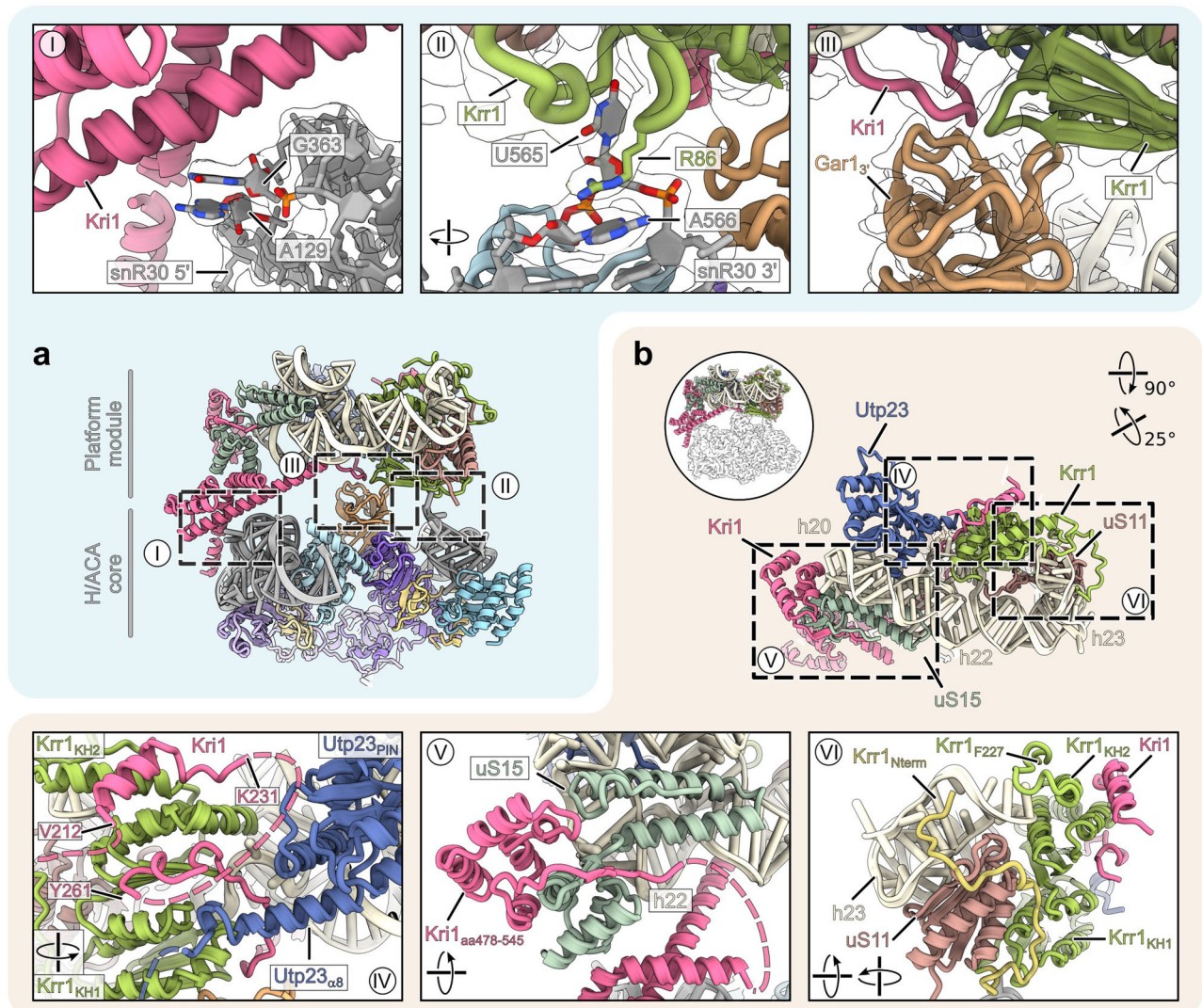

**Fig. 5 | Structural analysis of the platform module connected to the H/ACA module.** Overviews in the middle highlight the detailed views in (**a**, I–III) and (**b**, IV–VI). **a** The platform and H/ACA modules interact through three major contact sites. Interaction between Kri1 and the 5' half of snR30 (I). Interaction between Krr1 and nucleotides U565/A566 in the 3' hairpin of snR30 (II) and between the 3' Gar1, Kri1 and Krr1 (III). **b** The N-terminus of Utp23 interacts with Krr1 and Kri1 (IV). Kri1 and Krr1 chaperone the ribosomal proteins uS15 and uS11, respectively (V and VI).

pierces through the rRNA helices arrangement (Fig. 6d)[50]. Together, these findings indicate that the observed interactions not only support the overall architecture of the complex, but also contribute to the recruitment of the snoRNP to the 18S pre-rRNA.

### snR30 snoRNP and Utp23 chaperone 18S rRNA folding

As a result of the snR30 snoRNP complex formation, the 18S rRNA platform subdomain together with its ribosomal proteins uS11 and uS15 assemble as an independent unit in a close-to-mature conformation stabilized by Kri1 and Krr1. (Supplementary Fig. 8).

Additionally, the neighboring rRNA region comprising the rm1, rm2 motifs and its connecting helix-forming stretch are kept separated due to the snR30::18S rRNA hybrid formation and are prevented from forming the ES6 region prematurely (Fig. 4e, f). Notably, a specific role during platform formation can be assigned to Utp23. We observed a class in our Utp23-Krr1ΔC3 dataset, in which the C-terminus of Utp23 was bound to the snR30-ES6 three-way junction, however, its PIN domain was not yet interacting with the platform module and helix h20 adopted a premature conformation (Supplementary Figs. 6c and 9). Accordingly, integration of h20 apparently requires the

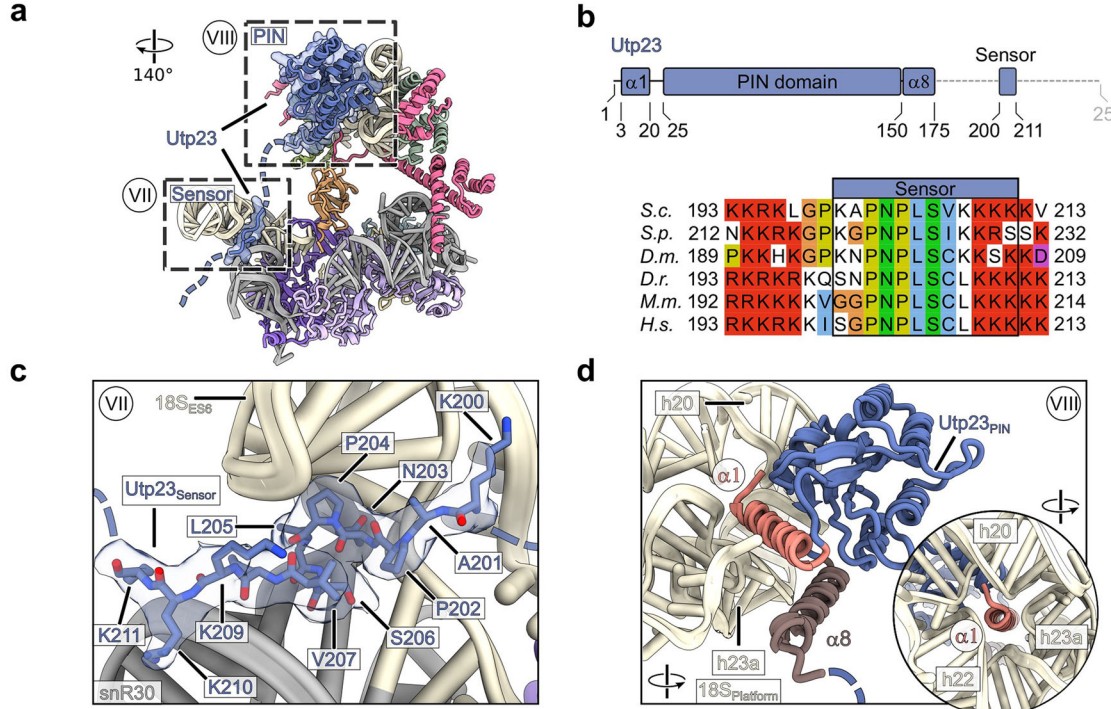

**Fig. 6 | Interaction of Utp23 with the snR30 snoRNP. a** Overview of the snR30 snoRNP, highlighting the detailed views of Utp23 shown in (**c** and **d**). **b** Domain organization of Utp23 (top panel) and multiple sequence alignment of the conserved region (sensor) recognizing the snR30-ES6 three-way junction. Sequences from *S.c. Saccharomyces cerevisiae*, *S.p. Schizosaccharomyces pombe*, *D.m. Drosophila melanogaster*, *D.r. Danio rerio*, *M.m. Mus musculus* and *H.s. Homo sapiens* were aligned with Clustal Omega and displayed using Jalview. **c** Molecular model and cryo-EM density map of the C-terminal sensor region of Utp23 in contact with 18S-ES6 and snR30. **d** Interaction of the Utp23 PIN domain and proximal α−helices 1 and 8 with the platform module rRNA.

sensing of the snR30:18S hybrid and subsequent binding of the Utp23 PIN domain to the platform module. In agreement with this role of Utp23, the deletion of the interacting sensor motif of the Utp23 C-terminus (aa202-211) results in a substantial growth defect[50]. Taken together, our results demonstrate that while individual components of the complex structure fulfill specific roles, their connectivity supports an overall architecture, which allows for an independent subdomain maturation including incorporation of uS11 and uS15.

### snR30 integration into the 90S intermediate

The snR30 snoRNP is released from the 18S rRNA in an early state to allow for a more compact 90S intermediate to form after the 3' minor domain of the 18S rRNA has been transcribed[9] (Supplementary Fig. 1a). The mutant Krr1ΔC3 allowed for the structural characterization of a later arrested state, in which the snoRNP interaction has remained intact and the failure of snR30 release resulted in stalling of the 90S maturation process. We therefore analyzed the snoRNP-bound 90S particles purified via the Krr1ΔC3 mutant to understand this arrest point (Fig. 2a, e). We observed several 90S intermediates, which all had in common an almost complete lack of the central domain including the platform domain (Supplementary Fig. 4f). The most matured state largely resembled the previously observed 90S state B2, yet also displayed a complementary gap, where usually the ES6 and platform module would be integrated (Figs. 7a, b and Supplementary Fig. 10a–c)[8]. Apparently, further 90S maturation is prevented by a failure of assembling the platform module into the 90S core, which is sterically hindered by the presence of the snR30 snoRNP and Kri1-Krr1-Utp23. Curiously, we observed another Krr1 molecule at its authentic position in the stalled 90S core (Supplementary Fig. 10b). Moreover, Kri1 and Utp23 block a variety of essential platform-90S interactions, including the binding site of uS15 to the C-terminus of Rrp7 by Kri1. Failure of Rrp7 recruitment to the 90S leads to a lack of UTP-C and consequently Rrp5 and Rok1, the helicase implicated in snR30 release

(see above and Supplementary Fig. 10c). Thus, the absence of UTP-C and Rrp5-Rok1 in our stalled 90S due to the Krr1 PBM mutation can explain why the inability of snR30 to dissociate from the 90S causes an early 90S biogenesis defect at the 18S rRNA platform region. Notably, the other 18S domains are assembled into this compact 90S particle, supporting the idea that they mature and assemble independently of each other (Fig. 7b, c).

### Discussion

Our study uncovered how the only essential H/ACA snoRNP in yeast, snR30, chaperones early 90S formation, which is directly coupled with the deposition of uS11 and uS15 via an assembly factor network comprising Kri1, Krr1 and Utp23. We found that Krr1 plays a crucial role in the release of this interwoven snR30 snoRNP structure. While we cannot explain the molecular basis of this release reaction yet, we took advantage of the release deficient mutant to identify the rRNA bound snR30 snoRNP and its connected stalled 90S pre-ribosome. To our surprise, we found the H/ACA snoRNP core forming a bipartite structure together with the 18S rRNA platform module, for which the RNP in concert with the AFs Krr1, Kri1 and Utp23 appears to play a crucial role in facilitating its independent folding (Fig. 7c).

Our data suggest several principles characterizing the snR30 snoRNP guided assembly of the 18S rRNA ES6 and the adjacent 40S platform region: (i) snR30 functions within a structurally defined complex that mediates independent 18S subdomain folding and concurrent delivery of ribosomal proteins, while also supporting recruitment of the snoRNP to and interaction with its 18S-ES6 target region; (ii) the close-to-mature conformation of the platform subdomain is stabilized by the interaction with the snoRNP and its associated factors (such as Kri1 and Utp23), while nearby rRNA regions, which could potentially interfere with the productive folding of this subdomain, are kept isolated and neutralized in an immature conformation by their participation in snR30::18S RNA hybrid formation; (iii) the complex

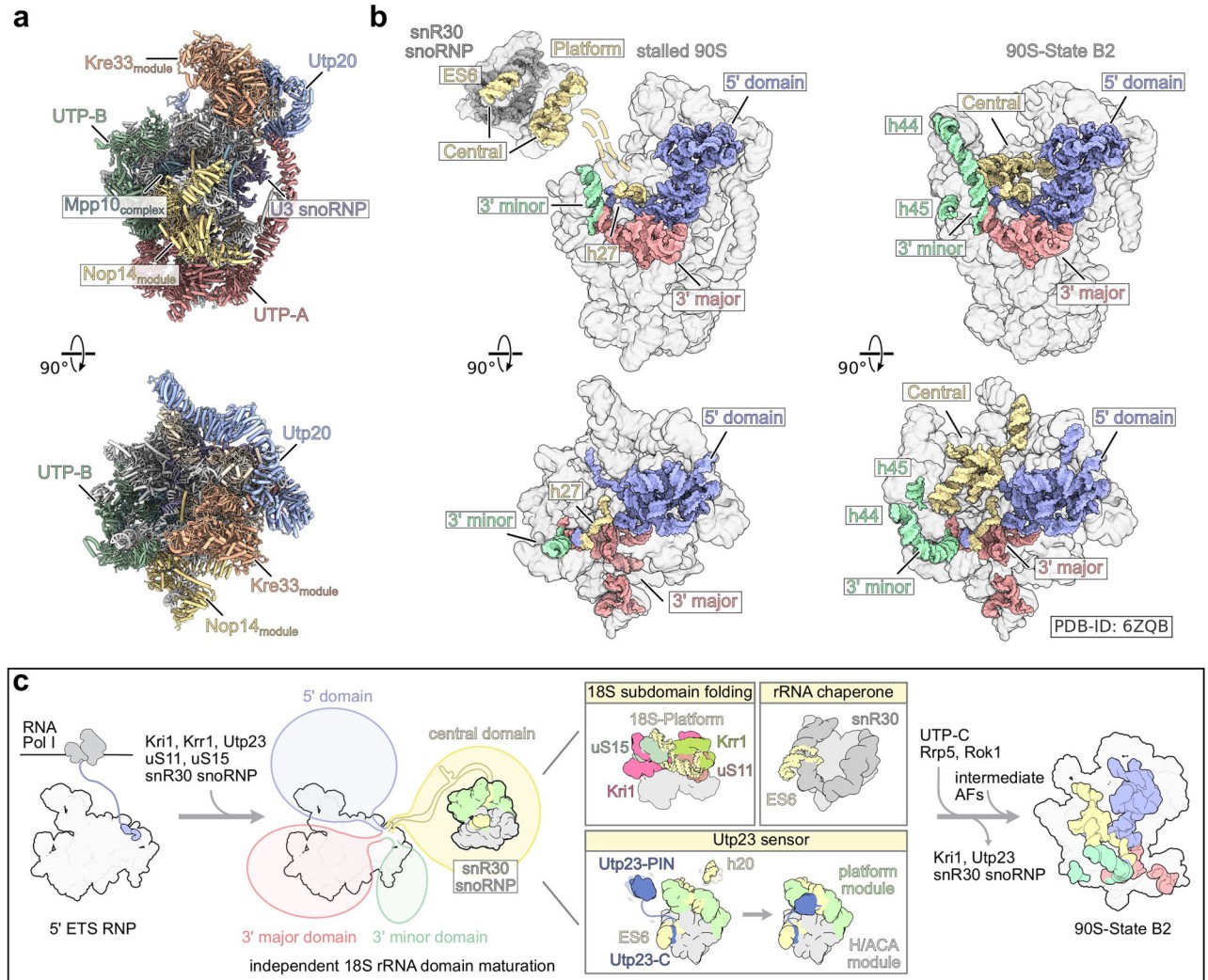

**Fig. 7 | Structure of the stalled Utp23-Krr1ΔC3 90S particle. a** Molecular model of the stalled 90S particle purified via Utp23-Krr1ΔC3 depicted in two orientations. Assembly factors and 90S biogenesis modules are labeled and colored.
**b** Transparent surface view of the stalled 90S particle and the snR30 snoRNP (left) and comparison with the 90S state B2 (right, PDB-ID: 6ZQB). The connection between the stalled 90S and the snR30 snoRNP are indicated with dashed lines. 18S rRNA domains are labeled and shown as coloured surfaces. **c** Proposed model illustrating early 90S pre-ribosome maturation steps, including the different functional roles of the snR30 snoRNP, which contributes to 18S central domain maturation.

between the snR30 H/ACA core and the 18S folding platform module sets the stage for 90S integration by coordinating the assembly factor Krr1, which facilitates 90S binding and the required release from the snoRNP core. Successful integration of the platform module, supported by an intact C-terminus of Krr1, could signal the completion of the rRNA chaperoning function of snR30 and thus induce the release (Fig. 7c). Similar principles have been observed in ribosome biogenesis before, however, usually mediated by AFs. One example is the independent assembly of the 5S RNP before integration into the nascent pre-60S subunit, where it is first kept in a rotated immature conformation before full integration[51,52]. Another example is the assembly and positioning of the L1 stalk region above the premature central protuberance of the pre-60S subunit before adopting its mature position at the ribosomal E-site[53].

Apparently, the ES6 and 40S platform region realize the same principles of an independent subdomain folding process by its interaction with the snR30 snoRNP, which takes place within a higher order structure that stabilizes an immature conformation of a potentially interfering neighboring region. With this unexpected finding, it will be interesting to explore to what extent the numerous other snoRNPs evolved beyond their original function of rRNA modification in order to support the assembly of ribosomal subunits.

## Methods

### Cloning and yeast transformation
Plasmids were constructed using standard procedures, and are detailed in Supplementary Table 2. DNA fragments were amplified using Phusion High-Fidelity DNA Polymerase (NEB), and restriction digests were carried out using enzymes from NEB and Thermo Fischer Scientific. Ligations were transformed into chemically competent E. coli DH5α. Point mutations were introduced by PCR reactions with primers harboring the desired mutation. DNA sequencing (Eurofins MWG-Operon; Ebersberg, Germany) was performed to verify the intact DNA sequence of the inserted fragment.

Yeast cells were transformed according to the lithium acetate standard method described in ref. 54.

### Generation of yeast strains
All yeast strains used in this study are listed in Supplementary Table 3. Gene disruption and genomic tagging was described before[55,56].

Correct genomic alterations were verified by colony PCR, detection of tagged gene variants by western blot and phenotype selection.

## Media and culture conditions

Yeast cells were grown under standard laboratory conditions (YPD, 30 °C). Cells were harvested during logarithmic growth phase. Yeast cells expressing a genomically protA-TEV-tagged Utp23 variant were transformed with a plasmid encoding Krr1ΔC3-FLAG under control of the endogenous promoter. Cells were grown in SDC medium, shifted to YPD medium and harvested after 6 h of growth.

YPD: 1% (w/v) yeast extract (MP), 2% (w/v) glucose (Merck), 2% (w/v) Bacto TM peptone (BD), pH 5.5

SDC-XY: 2% (w/v) glucose (Merck), 0.67% (w/v) yeast nitrogen base without amino acids, complemented with amino acids lacking XY (CSM drop-out, Formedium), pH 5.5

Yeast cells carrying a plasmid, on which the gene of interest was fused to a galactose inducible promoter, were grown in SDC medium. This selective preculture was used to inoculate 1 L of selective SRC medium ON. -16 h later, protein expression was induced by adding 1 L 2xYPG to a final galactose concentration of 2%. Cells were harvested after 6 h of galactose induction

SRC-XY: 2% (w/v) raffinose (MP), 0.67% (w/v) yeast nitrogen base without amino acids, complemented with amino acids lacking XY (CSM drop-out, Formedium), pH 5.5

YPG: 1% (w/v) yeast extract (MP), 2% (w/v) galactose (Sigma), 2% (w/v) BactoTM peptone (BD), pH 5.5

## Complementation assay and growth analysis of *krr1* mutants

To analyze cell growth, yeast cells were plated in 10-fold serial dilutions on specified plates, and the plates were then incubated for 2 or 3 days at the designated temperatures. Complementation was tested on SDC + FOA plates with growth on SDC plates as control. Growth of viable *krr1* mutants was tested on YPD at 23 °C, 30 °C and 37 °C. Lethal *krr1* gene variants fused to a galactose inducible promoter were analyzed for dominant-negative growth defects by plating cells on SGC plates with growth on SDC plates as control.

## Affinity purification of pre-ribosomal particles

Strains expressing the tagged bait proteins were collected during the logarithmic growth phase, subjected to a cold $H_2O$ rinse, resuspended in Buffer A (60 mM Tris-HCl, pH 8.0, 50 mM NaCl, 40 mM KCl, 2 mM $MgCl_2$, 1 mM dithiothreitol, 5% (v/v) glycerol, 0.1% (v/v) IGEPAL CA-630, SIGMAFAST complete protease inhibitor cocktail (Sigma–Aldrich)), frozen in liquid nitrogen, and stored at −80 °C until further processing. The harvested cells were mechanically disrupted using a cryogenic cell mill (Retsch MM400). The resulting lysate was centrifuged twice (10 min at 4600 × g, followed by 20 min at 35,000 × g, 4 °C). Subsequently, yeast extracts were incubated with pre-equilibrated immunoglobulin G (IgG) Sepharose 6 Fast Flow beads (GE Healthcare) for a minimum of 2 h. After three washes with 5 mL of Buffer B (60 mM Tris-HCl, pH 8.0, 50 mM NaCl, 40 mM KCl, 2 mM $MgCl_2$, 1 mM dithiothreitol, 5% (v/v) glycerol, 0.01% (v/v) IGE-PAL CA-630), the mixture was transferred to Mobicol columns (Mobitech), and bound proteins were eluted via TEV protease cleavage at 16 °C for 2 h. In a subsequent affinity purification step, TEV protease eluates were exposed to FLAG agarose beads (ANTI-FLAG M2 Affinity Gel, Sigma–Aldrich) at 4 °C for 2 h. The FLAG beads were washed once with 5 mL of Buffer B, and bound proteins were eluted with FLAG peptide (concentration 100 μg/ml, 1 h, 4 °C). The final eluates were separated on 4–12% polyacrylamide gels (NUPAGE, Invitrogen) using colloidal Coomassie staining and/or further analyzed with mass spectrometry or used to isolate and analyze RNA content of the sample. FLAG beads of samples that were subsequently analyzed by cryo-EM were washed and eluted in Buffer C (60 mM Tris-HCl, pH 8.0, 50 mM NaCl, 40 mM KCl, 2 mM $MgCl_2$,

1 mM dithiothreitol, 2% (v/v) glycerol, 0.05% (v/v) Octaethylene Glycol Monododecyl Ether).

## Mass spectrometry

FingerPrints proteomics at the University of Dundee, UK, conducted semiquantitative mass spectrometry. Co-precipitating proteins were identified through 1D nLC-ESI-MS-MS, and the raw data was analyzed using MaxQuant software[57]. The resulting iBAQ values were normalized to Pwp2 (Figs. 1f and 2b) or the respective bait proteins (Fig. 3c).

## rRNA isolation and northern blot

For RNA extraction from final FLAG eluates and sucrose gradient fractions, the protocols described previously[58,59] was followed. All centrifugation steps were performed at 4 °C. In essence, Phenol-Chloroform-IAA was added, mixed by shaking for 1 minute, and then centrifuged for 5 min at maximum speed. The aqueous phase was combined with chloroform, mixed by shaking, and centrifuged again for 5 min at maximum speed. The resulting aqueous phase was mixed with three volumes of 100% ethanol and 1/10 volume of 3 M NaOAc pH 5.2, and RNA was precipitated for 2–3 h at −20 °C. The precipitated RNA was pelleted by centrifugation for 20 min at maximum speed, washed once with 70% ethanol, and air-dried before resuspension in water.

Subsequently, the extracted RNA was loaded and resolved on an 8% polyacrylamide gel (with 8 M urea) after denaturation with formaldehyde. RNA was then transferred onto positively charged nylon membranes (GE healthcare) and crosslinked with UV. For northern blotting, the following DNA nucleotide probes were 5′-end $^{32}$P-ATP radiolabeled using T4 PNK (NEB): 5′-TTATGGGACTTGTT-3′ (probe for U3[18]), 5′-TCACTCAGACATCCTAGG-3′ (probe for U14[60] and 5′-ATGTCTGCAGTATGGTTTTAC-3′ (probe for snR30[61]).

## Sucrose gradient fractionation of isolated pre-ribosomes

FLAG-eluates of split-tag tandem affinity purifications were directly transferred to a linear 10–40% (w/v) sucrose gradient (60 mM Tris-HCl (pH 8.0), 50 mM NaCl, 2 mM $MgCl_2$, 0.003% IGEPAL CA-630 and 1 mM DTT), and centrifuged for 16 h at 129,300 × g at 4 °C. After centrifugation, the sucrose gradient was fractioned and each fraction was equally split. One half was TCA-precipitated, proteins were resuspended in sample buffer and analyzed by SDS-PAGE followed by staining with colloidal Coomassie (Roti Blue, Roth). RNA of the other half was Phenol-Chloroform extracted and precipitated with ethanol for subsequent northern analysis as described before[8,62].

## Cryo-EM and image processing

A Vitrobot Mark IV (FEI Company) was used to apply the purified samples (3.5 μl) to freshly glow discharged R3/3 holy copper grids supported with 3 nm continuous carbon (Quantifoil). The samples were incubated for 45 s on the grids at 4 °C and 95% humidity, blotted for 3 s and plunge frozen in liquid ethane. The data was collected on a Titan Krios G3 (Thermo Fischer) operating at 300 keV and equipped with a K2 summit direct electron detector (Gatan). Movies (40 frames per movie) were collected with a pixel size of 1.045 Å/pixel and at a defocus range of −0.5 to 3.5 μm under low dose conditions with a total electron dosage between 46.4 and 50 e⁻/Å². Movie frames were dose-weighted, aligned, summed, and motion corrected using MotionCor2[63] and contrast transfer function (CTF) parameters were estimated with CTFFIND4[64].

For the Utp23-Krr1 wt sample 12,520 micrographs were collected and after manual inspection, 10,999 micrographs were imported into cryoSPARC[65]. Patch CTF Estimation was performed and particles were initially picked with Blob Picker, extracted with a pixel size of 2.090 Å/pixel and a box size of 250 × 250 pixel and used for 2D classification.

We did not obtained any potential 90S class averages for further processing but identified three distinct class averages which together contained in total 36,427 snoRNP particles. The particles were re-extracted and used for Topaz training[66], 2D classification, Ab-initio reconstruction and two rounds of heterogeneous refinement. We obtained one snR30 snoRNP class containing 48,050 particles. Two additional rounds of Topaz training combined with two rounds of heterogeneous refinement increased the number of good particles to 62,516. Undecimated particles were extracted (1.045 Å/pixel, box size 320 × 320 pixel) and used for Non-uniform Refinement and 3D Classification which resulted in two snR30 snoRNP classes with overall resolutions of 3.92 and 3.93 Å (see Supplementary Figs. 4a–c and 6a). In the case of the Utp23-Krr1ΔC3 sample, 10,796 micrographs were collected and after manually inspection, 10,053 micrographs were used for further processing. Particles were initially picked without reference using the Relion 3.1.3 Laplacian-of-Gaussian blob detection (Autopick)[67]. The particles were extracted with a pixel size of 2.090 Å/pixel and a boxsize of 250 × 250 pixel and particles were imported to cryoSPARC[65]. Three consecutive rounds of 2D classification were performed and we obtained 314,297 particles with clear 90S features and 74,965 particles of the snR30 snoRNP. For both complexes Ab-Initio Reconstruction and Homogeneous Refinement were performed. The 90S particles were reimported into Relion 3.1.3 and extracted with a pixel size of 3.135 Å/pixel (box size of 180 × 180 pixel). 3D Refinement (initial angular sampling of 1.8°) and subsequent 3D Classification without alignment (T8, E20) were performed for sorting of the particles. We obtained six distinct classes which mainly differed in the presence or absence of Utp20, the Kre33 module and the previous described Lcp5-Bfr2 complex[68]. In addition, Utp10 was found in different conformations. All states lacked the central domain of the 18S rRNA (with the exception of the nucleotides of h27, 1111–1136 nt), including ES6 and platform helices h20, h22 and h23 and the associated UTP-C module as well as ribosomal proteins eS1, eS27, uS11 and uS15. The most matured state contained 32,278 particles and was comparable to the previous characterized state B2 without the mentioned central domain and associated factors (stalled 90S). The corresponding particles were extracted with a pixel size 1.045 Å (box size of 500 × 500 pixel), imported into cryoSPARC and refined using Non-uniform Refinement together with Defocus and Global CTF Refinement. In order to increase the resolution of flexible parts of the 90S, Local Refinements were performed and a composite map was generated (see Supplementary Fig. 4d–f and 6d). During processing of the snR30 snoRNP population we realized that better results were obtained when also particles without clear 2D averages were included in the refinement. Therefore, we combined the 74,965 with 281,029 particles with unclear 2D classes and extracted them in Relion 3.1.3 with a pixel size of 2.090 Å/pixel (box size of 130 × 130 pixel). 3D refinement was performed with the obtained Ab-initio map as initial model following 3D classification with image alignment using an angular sampling interval of 7.5°. One clear snoRNP class was obtained, which was extracted with a pixel size of 1.045 Å and further processed in cryoSPARC. After Heterogeneous and Non-uniform Refinement we obtained a cryo-EM density map with an overall resolution of 3.47 Å. Nevertheless, the resolution was insufficient for model building of the snR30 snoRNA (see Supplementary Fig. 4d–f). To increase the particle number and resolution of the snR30 snoRNP population the micrographs were imported into cryoSPARC following Patch CTF Estimation. Particles were picked without reference using Blob picker, extracted with a pixel size of 2.09 Å/pixel. After 2D classification four snoRNP classes were selected, containing 60,780 particles in total, that were used for Topaz training. Through several rounds of heterogeneous refinement and Topaz training, a snR30 snoRNP class containing 618,743 particles was obtained. Particles were extracted undecimated (1.045 Å/pixel, box size 320 × 320 pixel) and used for Non-uniform Refinement, 3D Classification and Heterogeneous

Refinements as illustrated in Supplementary Fig. 5a. Two states were obtained one containing clear density for the Utp23 PIN domain (snR30 snoRNP Utp23-Krr1ΔC3 State1) and the other lacking the PIN domain at average resolutions of 2.89 Å and 3.18 Å, respectively. Local Refinements of the platform module and the 5′ and 3′ parts of the H/ACA module were performed for the State1 (Supplementary Fig. 6b, c).

## Model building and refinement

The molecular model of the 90S state B2 (PDB-ID: 6ZQB)[8] was used as an initial reference for the stalled 90S and fitted into the cryo-EM density map with ChimeraX[69]. The model of the missing central domain rRNA (between 624–1084 nt), parts of the 3′ minor domain (1676–1724 nt, 1768–1790 nt) as well as interacting protein chains: ribosomal proteins eS1, eS27, uS15, uS11, the UTP-C complex (Utp22, Rrp7), the double WD-propeller of Utp13 (aa1-645) and the C-terminus of Krr1 (aa247-307) were removed from the model. The individual chains were rigid body fitted with Phenix[70]. Alphafold[71] and Alphafold multimer (v2.3.1) (https://doi.org/10.1101/2021.10.04.463034) were used to assist model building and refinement and the model was manually adjusted with Coot[72].

In case of the Utp23-Krr1ΔC3 snR30 snoRNP State 1, the *Saccharomyces cerevisiae* Cbf5-Nop10-Gar1 complex (PDB-ID: 3U28)[25] and the human telomerase H/ACA RNP (PDB-ID: 7TRC)[73] were used as initial models for the H/ACA core and the crystal structure of the Utp23 PIN domain (PDB-ID: 4MJ7)[50] and the models of Krr1, uS11, uS15 and the platform 18S rRNA from PDB-ID: 6ZQB[8] as initial references for the platform module. Extra density around the Krr1 KH domains and uS15 were identified as Kri1 assisted by AlphaFold multimer (v2.3.1) (https://doi.org/10.1101/2021.10.04.463034). All initial models were rigid body fitted into the cryo-EM density and manually adjusted with Coot[72]. The model of the snR30 was build de novo using the conserved H and ACA box models of the human telomerase H/ACA RNP (PDB-ID: 7TRC)[73] as a starting point.

For the model of Utp23-Krr1ΔC3 snR30 snoRNP State2 the rigid body fitted model of State 1 was used as initial reference. The models for the Utp23-PIN domain (aa1-175) and for 18S rRNA h22 were removed and the chains were manually adjusted with Coot. Finally, all models were real space refinement with Phenix[70].

Visualization of molecular models and cryo-EM density maps was performed with ChimeraX[69].

## Reporting summary

Further information on research design is available in the Nature Portfolio Reporting Summary linked to this article.

## Data availability

The data supporting the findings of this study are available from the corresponding authors upon request. Mass-spectrometry raw data generated in this study has been deposited to the ProteomeXchange Consortium via the PRIDE[74] partner repository. The datasets are available via the PRIDE website under accession codes: PXD054031 (Dataset 1 Fig. 1f), PXD054032 (Dataset 2 Fig. 2b), PXD054034 (Dataset 3 Fig. 2d), PXD054035 (Dataset 4 Supplementary Fig. 3c). Cryo-EM maps and molecular models generated in this study have been deposited to the Electron Microscopy Data Bank (EMDB) and the Protein Data Bank (PDB), respectively under accession codes: snR30 snoRNP Utp23-Krr1ΔC3 State1 (PDB-9G25, EMD-50964, EMD-50958, EMD-50959, EMD-50960 and EMD-50961) snR30 snoRNP Utp23-Krr1ΔC3 State2 (EMD-50968 and PDB-9G28), Stalled 90S Utp23-Krr1ΔC3 (PDB-9G33, EMD-50991, EMD-50647, EMD-50648, EMD-50649, EMD-50650, EMD-50651, EMD-50652, EMD-50653, EMD-50654), snR30 snoRNP Utp23-Krr1 wt Class1 (EMD-50967) and snR30 snoRNP Utp23-Krr1 wt Class2 (EMD-50969). Source data are provided with this paper.

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

## Acknowledgements

We thank Susanne Rieder and Charlotte Ungewickell for technical sup-port. The study was supported by DFG grant HU363/12-1 and ERC grant ADG 741781 GLOWSOME (E.H.) and ERC grant ADG 885711 HumanRi-bogenesis (R.B.).

## Author contributions

P.F., M.T., E.H., and R.B. designed the study. P.F. designed and generated all yeast strains and mutants, and P.F. and M.K. performed growth analysis of these strains. P.F., B.L., and M.K. isolated pre-ribosomal particles for biochemical and cryo-EM analysis. B.L. performed sucrose gradient centrifugation with subsequent SDS-PAGE and Northern blot analysis. D.F. screened the samples by negative stain EM analysis. O.B. performed cryo-EM data collection. M.T. prepared the cryo-EM grids and processed the cryo-EM data. M.T. built and refined the molecular models with the help of T.D. Structures were analyzed and interpreted by M.T. and R.B. The Supplementary Movie was created by T.D. The manuscript was written by P.F., M.T., R.B., and E.H., and all authors commented on the manuscript.

## Funding

## Competing interests

The authors declare no competing interests.
