## [Transparent Peer Review file · Nature Communications]

H/ACA snR30 snoRNP guides independent 18S rRNA subdomain formation

Corresponding Author: Professor Roland Beckmann

Version 0:

Reviewer comments:

Reviewer #1

(Remarks to the Author)

Fischer et al. present a comprehensive study investigating the role of snR30 in promoting small ribosomal subunit assembly. Although snR30 was identified as a crucial component in ribosome assembly more than 30 years ago, its precise molecular interactions with 40S assembly intermediates have remained unclear, despite prior mapping and crosslinking studies defining certain aspects of its biological function. In this paper, the authors explore the link between snR30 and the ribosome biogenesis factor (RBF) Krr1, demonstrating that a C-terminal truncation of Krr1 is essential for growth and leads to a dominant-negative phenotype upon overexpression.

Notably, the authors observe that snR30 accumulates specifically in late 90S intermediates, suggesting that the absence of the Krr1 C-terminal tail effectively "traps" snR30 on the pre-rRNA. Structural analysis of these intermediates reveals that snR30 forms a distinct structure containing not only expected small nucleolar ribonucleoprotein (snoRNP) components but also a complex comprising rRNA helices, the ribosomal proteins uS11 and uS15, and the RBFs Kri1, Utp23, and Krr1. This combined structure appears as a tethered "satellite" to the main body of the 90S complex.

These findings provide a structural basis for the function of snR30 in 40S biogenesis, while also revealing, for the first time to my knowledge, that certain ribosomal assembly events occur within separate structured modules, distinct from the rRNP core structure under assembly. This study will undoubtedly capture the interest of researchers in ribosome biogenesis and is likely to appeal to a broader audience interested in RNA biology. Additionally, this work is significant for another reason: many structural studies on ribosome assembly to date, especially those focusing on pre-60S intermediates, tend to overlook large, dynamically associated rRNA domains if they cannot be separately resolved in cryo-EM reconstructions. This study suggests that new strategies should be developed to visualize dynamically tethered, partially formed rRNA domains.

I fully support the publication of this paper in Nature Communications, with the following specific suggestions:

1. Clarifying the structure introduction in Fig. 3:

My primary critique concerns how the snRNP/rRNA structural module is introduced in Figure 3. The initial reference to Figure 3 describes the structure as the "snR30 snoRNP" complex, which is confusing since a cursory look at the figure indicates a more complex structure. The text then describes additional elements associated with the snoRNP core but does not immediately clarify that the structure is tethered to the 90S core. I recommend that the authors introduce the overall structure upfront in the text, explicitly stating that it comprises a 90S core with a tethered snoRNP/rRNA domain. Including a schematic (e.g., a simplified cartoon similar to Figure 6b) would be beneficial. The detailed description of the 90S structure can be deferred, but establishing context early is crucial to allow a broad readership to understand what is going on.

2. Improving clarity in Fig. panels 3c and 3d:

Panels 3c and 3d are currently not very effective in illustrating which parts of the rRNA are present in the tethered structure. Displaying the mature secondary structure of ES6 is misleading, as it differs from the structure required to bind snR30. The authors should incorporate information similar to Figure 3 in their reference 46, highlighting the alternative structure of ES6 observed in the reconstruction and how it must change to achieve the mature 40S conformation. Additionally, the coloring scheme for the 3' and 5' hairpins in panel 3d should be consistent with that used in panels 3a and 3b. Without reading the legends, one might mistakenly assume that the grey RNA segments correspond to those shown in panel 3c. Consistently coloring snR30 as in panel ES7a would facilitate interpretation of the model. Overall, the reader is required to put in too much effort to grasp what should be the paper's most exciting reveal. The same coloring scheme would also be beneficial in Figs. 4 and 5.

3. Clarifying rRNA Linkages:

It would be beneficial to know the number of rRNA residues linking the 90S core and the isolated domain. A secondary structure diagram of the rRNA, showing which features (if any) of the central domain are visible within the 90S structure, would help convey the approximate length of the tether between the isolated domain and the core. Along these lines, have the authors attempted to pair 90S and snoRNP particles in their dataset (i.e., by re-extracting snoRNP particles using larger box sizes)? This additional analysis could provide further insights into how dynamic the link between the snRNP and the 90S is.

Other comments:

There are several instances of awkward wording throughout the manuscript. Two specific examples include:

Line 34: The sentence starting with "Apparently,..." is confusing and should be rephrased to convey the intended meaning more clearly.

Line 50: The sentence beginning with "Only..." is unclear due to improper word order.

Reviewer #2

(Remarks to the Author)

We participated in the Nature Communications Early Researcher Review program and the comments below represent our joint review.

In this manuscript Fisher et al characterize the structure and function of snR30, the only essential H/ACA small nucleolar (sno)RNA in yeast, which is well conserved across eukaryotes. Assembly of the small ribosomal subunit requires ~70 protein assembly factors as well as several snoRNAs that coordinate the modular assembly of the 18S pre-rRNA, but the function of many of these factors remains underexplored. Previous structural studies revealed that assembly factor Krr1 associates with a region of the 18S rRNA that is modified by snR30. To probe the function of Krr1, the authors made a series of truncations of Krr1 in yeast and discovered that a proline-rich motif from the C-terminus is critical for ribosome assembly and required for the release of the snR30 snoRNP from the pre-90S. Next the authors used a split-bait approach to isolate pre-90S particles with either WT-Krr1 or Krr1 lacking part of the C-terminus for structural and functional analysis. WT-Krr1 resulted in the isolation of early pre-90S particles that could not be characterized structurally. Removal of the C-terminus resulted in the isolation of mid-stage pre-90S particles and yielded two well behaved particle populations by single particle cryo-EM, including a series of pre-90S particles and the snR30 snoRNP complex. While the snR30 snoRNP complex appears to be an isolated complex by 2D classes, it is flexibly tethered to the 90S via the platform module which contains part of the 18S rRNA in a close to mature conformation.

Overall, the authors present compelling genetic and structural data which reveals exciting new insight into the role that the h/ACA snR30 snoRNP plays in chaperoning an important subdomain of the small ribosomal subunit. We are supportive of publication once the following concerns are addressed:

- The majority of the figures in this manuscript are very small. Most figures fill less than half a page but contain numerous panels making the font size for the labels super small. It would be beneficial to increase the size of many of the figures (especially Figs. 1, 4, and 5) to full page figures to make the structures, labels, etc more legible. The same applies to many of the supplemental figures as well including S4 (this should be broken up into two figures) and S10.
- The order that the structures are presented in the text and figures is a bit confusing. It would be helpful to readers to move panels a and b from Figure 6 to Figure 3 as it is important to show the secondary structure cartoon of the 18S alongside both structures to make it clear which regions of the 18S associate with the snoRNP and which regions form the other regions of the pre-90S. This would also reinforce the message that the two structures are connected via a flexible a tether.
- The authors should refer to the Platform module as the "Krr1deltaC3 Platform Module" in the text and figures.
- In Figure 3 and Figure 5 please label the C-terminus of Krr1, to help readers visualize where the truncation begins.
- In Figure 4 panel a; please label the platform module and H/ACA core in the top left overview, also please clarify the rotation of the two models in relation to the overview.
- In Figure 4 panel c; the colors of Nh2p5' and Nh2p3' of inset III and IV respectively are colored opposite of the model in panel a. Please change labels and/or colors to make this match the model from panel a.
- In Figure 4 panel g; it appears that the comparison is being made between the top right and the two bottom panels to show the substrate binding regions, and the presence/arrangement of the top left panel (corresponding to inset V) is a different area of the H/ACA core and therefore not relevant to be arranged with the other three panels. Separating the panel of inset V from inset VI, 3HAY, and 8OUE would provide more clarity for the comparison being made.
- Panels c and d of Figure 5 containing the model of the platform module and the H/ACA core would be more effective as panels a and b (and moving current panel a and b to become c and d) as this is a match to the overview presented inside the circle and then the following panels being closer looks at individual sections of the module.

- Can the authors see any hints at an association between the snoRNP and the rest of the pre-90S. If you expand the box size can you see any classes with both the pre-90S and snoRNP modules? Have the authors tried focused refinement around the area where the snoRNP should extend from the rest of the pre-90S? How many bases of the 18S are disordered/invisible between the connection points?

Reviewer #3

(Remarks to the Author)

Version 1:

Reviewer comments:

Reviewer #2

(Remarks to the Author)

We participated in the Nature Communications Early Researcher Review program and the comments below represent our joint review.

The authors have done an excellent job addressing our previous concerns. We appreciate the updated figures and are fully supportive of publishing this manuscript in Nature Communications. Congratulations to the authors on this nice story revealing the structure and function of the h/ACA snR30 snoRNP in ribosome assembly.

Reviewer #3

(Remarks to the Author)

REVIEWER COMMENTS

Reviewer #1 (Remarks to the Author):

Fischer et al. present a comprehensive study investigating the role of snR30 in promoting small ribosomal subunit assembly. Although snR30 was identified as a crucial component in ribosome assembly more than 30 years ago, its precise molecular interactions with 40S assembly intermediates have remained unclear, despite prior mapping and crosslinking studies defining certain aspects of its biological function. In this paper, the authors explore the link between snR30 and the ribosome biogenesis factor (RBF) Krr1, demonstrating that a C-terminal truncation of Krr1 is essential for growth and leads to a dominant-negative phenotype upon overexpression. Notably, the authors observe that snR30 accumulates specifically in late 90S intermediates, suggesting that the absence of the Krr1 C-terminal tail effectively "traps" snR30 on the pre-rRNA. Structural analysis of these intermediates reveals that snR30 forms a distinct structure containing not only expected small nucleolar ribonucleoprotein (snoRNP) components but also a complex comprising rRNA helices, the ribosomal proteins uS11 and uS15, and the RBFs Kri1, Utp23, and Krr1. This combined structure appears as a tethered "satellite" to the main body of the 90S complex. These findings provide a structural basis for the function of snR30 in 40S biogenesis, while also revealing, for the first time to my knowledge, that certain ribosomal assembly events occur within separate structured modules, distinct from the rRNP core structure under assembly. This study will undoubtedly capture the interest of researchers in ribosome biogenesis and is likely to appeal to a broader audience interested in RNA biology. Additionally, this work is significant for another reason: many structural studies on ribosome assembly to date, especially those focusing on pre-60S intermediates, tend to overlook large, dynamically associated rRNA domains if they cannot be separately resolved in cryo-EM reconstructions. This study suggests that new strategies should be developed to visualize dynamically tethered, partially formed rRNA domains.

We thank the reviewer for the positive evaluation of our work. Our responses to the points raised are shown in blue below each point.

I fully support the publication of this paper in Nature Communications, with the following specific suggestions:

1. Clarifying the structure introduction in Fig. 3:

My primary critique concerns how the snRNP/rRNA structural module is introduced in Figure 3. The initial reference to Figure 3 describes the structure as the "snR30 snoRNP" complex, which is confusing since a cursory look at the figure indicates a more complex structure. The text then describes additional elements associated with the snoRNP core but does not immediately clarify that the structure is tethered to the 90S core. I recommend that the authors introduce the overall structure upfront in the text, explicitly stating that it comprises a 90S core with a tethered snoRNP/rRNA domain. Including a schematic (e.g., a simplified cartoon similar to Figure 6b) would be beneficial. The detailed description of the 90S structure can be deferred, but establishing context early is crucial to allow a broad readership to understand what is going on.

We agree with the reviewer regarding this point. The snR30 snoRNP is the complete bipartite structure containing the H/ACA core and the associated platform module. We have clarified this in the revised manuscript. As suggested by the reviewer we have included a snR30-90S cartoon up front as a new Figure 3a. We have also included 2D class averages of the snR30 snoRNP and 90S classes and 2D class averages of the snR30 snoRNP with an enlarged box size. A fuzzy density can be seen near the snR30 snoRNP, likely representing the 90S particle. Together, the schematic overview and class averages help to clarify our main finding that the H/ACA snoRNP is connected to the 90S via the 18S central subdomain.

See also Reviewer #2/3 and Reviewer 1 (point 3)

2. Improving clarity in Fig. panels 3c and 3d:

Panels 3c and 3d are currently not very effective in illustrating which parts of the rRNA are present in the tethered structure. Displaying the mature secondary structure of ES6 is misleading, as it differs from the structure required to bind snR30. The authors should incorporate information similar to Figure 3 in their reference 46, highlighting the alternative structure of ES6 observed in the reconstruction and how it must change to achieve the mature 40S conformation. Additionally, the coloring scheme for the 3' and 5' hairpins in panel 3d should be consistent with that used in panels 3a and 3b. Without reading the legends, one might mistakenly assume that the grey RNA segments correspond to those shown in panel 3c. Consistently coloring snR30 as in panel ES7a would facilitate interpretation of the model. Overall, the reader is required to put in too much effort to grasp what should be the paper's most exciting reveal. The same coloring scheme would also be beneficial in Figs. 4 and 5.

We agree in principle with the reviewer but decided not to use the snoRNA color scheme of former Figure 3d in former Figures 3a-b. Overall, there are limited color options available for the many factors and therefore additional colors for the different snR30 elements would probably be even more confusing. Instead, we now include an additional panel focusing on the structure of snR30 with its 5', 3' and internal hairpins as well as the H and ACA boxes and together with the 18S rRNA contained in the snR30 snoRNP (new Fig. 3d). We show this together with the snR30 secondary structure, colored in the same way to make it clearer. Since the orientation of the snoRNP is identical in Fig. 3c and Fig. 3d, we hope that it is easier now to understand without consulting the legend. For Figures 4 and 5, we have decided to use grey for the whole snR30 for clarity. We have also added a Supplementary Movie 1 showing the bipartite organization of the non-canonical snR30 snoRNP.

3. Clarifying rRNA Linkages:

It would be beneficial to know the number of rRNA residues linking the 90S core and the isolated domain. A secondary structure diagram of the rRNA, showing which features (if any) of the central domain are visible within the 90S structure, would help convey the approximate length of the tether between the isolated domain and the core. Along these lines, have the authors attempted to pair 90S and snoRNP particles in their dataset (i.e., by re-extracting snoRNP particles using larger box sizes)? This additional analysis could provide further insights into how dynamic the link between the snRNP and the 90S is.

A suggested, we have included an additional secondary structure scheme as a new panel Figure 3f. We illustrate the rRNA elements visible in the snR30 snoRNP and the connecting regions within the 90S particle. Residues connecting the rRNA in the snR30 snoRNP and the rRNA within the 90S, as well as regions of the central domain/ES6 that are unstructured or not visible in the snR30 structure, are shown as dashed lines with residues labelled. This should certainly help to convey the approximate length of the tether between the isolated domain and the core.

We have indeed tried to identify 90S snoRNP particles in our data set by increasing the box size. However, the connection between the two structures appears to be too flexible for structural studies. We have included 2D class averages of the snR30 snoRNP, the 90S particle and the snR30 snoRNP extracted with an increased box size as a new Figure 3a. For the different 2D class averages, a large but fuzzy density appears next to the snoRNP, which very likely corresponds to the bound 90S particle, together indicating a high degree of orientational flexibility. In agreement with the available rRNA linker length a limited distance between the 90S core and the snoRNP can be observed, however, no further conclusions on the dynamic link between the snRNP and the 90S are possible. It is apparently too dynamic as to allow for any joint 3D refinement.

See also the last point of Reviewers 2/3.

Other comments:

There are several instances of awkward wording throughout the manuscript. Two specific examples include:

Line 34: The sentence starting with "Apparently,..." is confusing and should be rephrased to convey the intended meaning more clearly.

Line 50: The sentence beginning with "Only..." is unclear due to improper word order.

We have improved the wording accordingly (e.g. lines 32, 89, 92, 172, 200, 295, 337, 347). Changes in the revised manuscript are highlighted in red.

Reviewer #2 (Remarks to the Author):

We participated in the Nature Communications Early Researcher Review program and the comments below represent our joint review.

In this manuscript Fisher et al characterize the structure and function of snR30, the only essential H/ACA small nucleolar (sno)RNA in yeast, which is well conserved across eukaryotes. Assembly of the small ribosomal subunit requires ~70 protein assembly factors as well as several snoRNAs that coordinate the modular assembly of the 18S pre-rRNA, but the function of many of these factors remains underexplored. Previous structural studies revealed that assembly factor Krr1 associates with a region of the 18S rRNA that is modified by snR30. To probe the function of Krr1, the authors made a series of truncations of Krr1 in yeast and discovered that a proline-rich motif from the C-terminus is critical for ribosome assembly and required for the release of the snR30 snoRNP from the pre-90S. Next the authors used a split-bait approach to isolate pre-90S particles with either WT-Krr1 or Krr1 lacking part of the C-terminus for structural and functional analysis. WT-Krr1 resulted in the isolation of early pre-90S particles that could not be characterized structurally. Removal of the C-terminus resulted in the isolation of mid-stage pre-90S particles and yielded two well behaved particle populations by single particle cryo-EM, including a series of pre-90S particles and the snR30 snoRNP complex. While the snR30 snoRNP complex appears to be an isolated complex by 2D classes, it is flexibly tethered to the 90S via the platform module which contains part of the 18S rRNA in a close to mature conformation.

We thank the reviewers for their positive feedback.

Overall, the authors present compelling genetic and structural data which reveals exciting new insight into the role that the h/ACA snR30 snoRNP plays in chaperoning an important subdomain of the small ribosomal subunit. We are supportive of publication once the following concerns are addressed:

- The majority of the figures in this manuscript are very small. Most figures fill less than half a page but contain numerous panels making the font size for the labels super small. It would be beneficial to increase the size of many of the figures (especially Figs. 1, 4, and 5) to full page figures to make the structures, labels, etc more legible. The same applies to many of the supplemental figures as well including S4 (this should be broken up into two figures) and S10.

We followed the reviewers suggestions and increased the size (Fig. 1, 3-6, S1, S3, S4, S5, S7, S8 and S10) and rearranged Figure 5 into the new Figures 5 and 6.

- The order that the structures are presented in the text and figures is a bit confusing. It would be helpful to readers to move panels a and b from Figure 6 to Figure 3 as it is important to show the secondary structure cartoon of the 18S alongside both structures to make it clear which regions of the 18S associate with the snoRNP and which regions form the other regions of the pre-90S. This would also reinforce the message that the two structures are connected via a flexible a tether.

The main focus of the manuscript is the novel structure of the snR30 snoRNP and we would like to keep the order of the results starting with snR30 itself. We agree that it is helpful for the reader that the snoRNP-90S connection is highlighted in Figure 3. Therefore, we have followed a similar suggestion from reviewer 1 and included a scheme of the snR30-90S complex into Figure 3, as well as other panels, to highlight the connection between the snR30 snoRNP and 18S central domain.

- The authors should refer to the Platform module as the “Krr1deltaC3 Platform Module” in the text and figures.

We prefer to keep the term 'platform module' in the figures. The platform module as outlined in our manuscript is defined as the 18S subdomain containing the 18S platform, complexed with uS11, uS15, Krr1, Kri1 and Utp23. The module can be identified both in the wt and Krr1ΔC3 purification. Nevertheless, we refer to the Krr1ΔC3 platform module in the text when it is first introduced (line 233).

- In Figure 3 and Figure 5 please label the C-terminus of Krr1, to help readers visualize where the truncation begins.

The Krr1ΔC3 mutant construct ends with V247 and the corresponding Krr1 model ends at position F227, before the applied truncations. We have labelled the last modelled C-terminal aa with Krr1 F227 in Figures 3, 5 and Supplementary Figure 9. In addition, we now mention the boundaries of the Krr1 model in the manuscript (line 214-216).

- In Figure 4 panel a; please label the platform module and H/ACA core in the top left overview, also please clarify the rotation of the two models in relation to the overview.

As suggested, the platform module and the H/ACA core are now labelled in the new Figure 4. The rotations of the two model figures relative to the overview have been indicated by the corresponding angles. Figure 4 is now rearranged and shown as a full-page figure (see first comment of reviewers 2/3). The overview and the two views of the model are now shown as equally sized figures, which hopefully helps with orientation of the model views.

- In Figure 4 panel c; the colors of Nh2p5' and Nh2p3' of inset III and IV respectively are colored opposite of the model in panel a. Please change labels and/or colors to make this match the model from panel a.

Thanks for pointing this out. The coloring was correct, but we accidentally mixed up the labelling. This has been corrected in the revised version.

- In Figure 4 panel g; it appears that the comparison is being made between the top right and the two bottom panels to show the substrate binding regions, and the presence/arrangement of the top left panel (corresponding to inset V) is a different area of the H/ACA core and therefore not relevant to be arranged with the other three panels. Separating the panel of inset V from inset VI, 3HAY, and 8OUE would provide more clarity for the comparison being made.

In the previous Figure 4g, the 5' half was overlaid as a rigid body with the model of the snR30 3' half, as was done for the archaeal H/ACA RNP and human telomere H/ACA RNP structures. The comparison of the 5'-half with the 3'-half of the snR30 snoRNP (V) is somewhat relevant, as it would be the second Gar1 binding site of the snR30 snoRNP that is blocked by the unique structure of the 5'-hairpin. However, since we already show the missing Gar1 copy in Figure 4d, we have moved the panel to Supplementary Figure 7 for clarity.

- Panels c and d of Figure 5 containing the model of the platform module and the H/ACA core would be more effective as panels a and b (and moving current panel a and b to become c and d) as this is a match to the overview presented inside the circle and then the following panels being closer looks at individual sections of the module.

We have rearranged Figure 5 to show the connection between the platform and the H/ACA core as a new Figure 5a and the details of the platform module as 5b. The following details of the Utp23 interaction are now shown as a new Figure 6 (see also first comment Reviewer 2/3).

- Can the authors see any hints at an association between the snoRNP and the rest of the pre-90S. If you expand the box size can you see any classes with both the pre-90S and snoRNP modules? Have the authors tried focused refinement around the area where the snoRNP should extend from the rest of the pre-90S? How many bases of the 18S are disordered/invisible between the connection points?

The point made by the reviewer is similar to one of the third comments from reviewer 1. We of course tried to expand the box size. However, the unresolved connecting areas between snR30 snoRNP and the stalled 90S particle are too large and therefore the 2 particles do not have a precise orientation to one another, which precludes focus refinement or multi body approaches. Nevertheless we include now 2D class averages of the snR30 particles with a larger box size. The classes illustrate that we frequently identify a large density 'cloud' in the vicinity of the snR30 particle which likely corresponds to the connected 90S particle.

In addition we include now a secondary structure scheme (Figure 3f) to better illustrate the missing connection between snR30 particles and 90S particles.

Reviewer #3 (Remarks to the Author):
